# Exploring the Therapeutic Potential of ^177^Lu-PSMA-617 in a Mouse Model of Prostate Cancer Bone Metastases

**DOI:** 10.3390/ijms26135970

**Published:** 2025-06-21

**Authors:** Cheng-Liang Peng, Chun-Tang Chen, I-Chung Tang

**Affiliations:** Department of Isotope Application Research, National Atomic Research Institute, Taoyuan 325207, Taiwan; ctchen@nari.org.tw (C.-T.C.); ictang@nari.org.tw (I.-C.T.)

**Keywords:** prostate cancer, bone metastasis, ^177^Lu-PSMA-617, targeted radionuclide therapy, preclinical model

## Abstract

Prostate cancer is the second leading cause of cancer-related death in men, with metastatic castration-resistant prostate cancer (mCRPC) and bone metastases representing a critical clinical challenge. Although radium-223 (Ra-223) is approved for treating mCRPC with bone metastases, its efficacy remains limited, necessitating the development of more effective therapies. This study investigates the therapeutic potential of ^177^Lu-PSMA-617, a PSMA-targeted radiopharmaceutical, in a murine model of prostate cancer bone metastases. To our knowledge, this is the first study to systematically evaluate ^177^Lu-PSMA-617 in an orthotopic bone metastatic prostate cancer model, providing a clinically relevant preclinical platform to assess both imaging and therapeutic performance. We conducted comprehensive preclinical evaluations, including synthesis, stability analysis, cell binding assays, nuclear imaging, in vivo biodistribution, pharmacokinetics, and antitumor efficacy. The synthesis of ^177^Lu-PSMA-617 demonstrated high radiochemical yield (99.2%), molar activity (25.5 GBq/μmol), and purity (>98%), indicating high product quality. Stability studies confirmed minimal release of free Lutetium-177, maintaining the compound’s integrity under physiological conditions. In vitro assays showed selective binding and internalization in PSMA-positive LNCaP prostate cancer cells, with negligible uptake in PSMA-negative PC-3 cells. In vivo biodistribution studies demonstrated efficient tumor targeting, with peak uptake in LNCaP tumors (23.31 ± 0.94 %IA/g) at 4 h post-injection. The radiopharmaceutical exhibited favorable pharmacokinetics, with high tumor-to-background ratios (tumor-to-blood, 434.4; tumor-to-muscle, 857.4). Therapeutic efficacy was confirmed by significant survival extension in treated mice (30.7% for 37 MBq and 53.8% for 111 MBq), with median survival times of 34 and 40 days, respectively, compared to 26 days in the control group. Radiation dosimetry analysis indicated a favorable safety profile with a calculated effective dose of 0.127 mSv/MBq. These findings highlight the novelty and translational relevance of using ^177^Lu-PSMA-617 in a clinically relevant bone metastasis model, reinforcing its potential as a dual-purpose agent for both targeted therapy and molecular imaging in advanced prostate cancer.

## 1. Introduction

Prostate cancer is the second leading cause of cancer-related mortality among men worldwide, following lung cancer, and predominantly affects the elderly population [1]. While the majority of cases exhibit an indolent course, certain subtypes display aggressive behavior, leading to poor clinical outcomes [2]. Current therapeutic strategies include surgery, radiation therapy, and endocrine therapy; however, effective treatment options remain limited, particularly for advanced-stage disease [3,4]. When prostate cancer progresses to metastatic castration-resistant prostate cancer (mCRPC) after developing resistance to androgen deprivation therapy (ADT), clinical management becomes increasingly challenging due to the scarcity of effective therapeutic options [5]. According to estimates from the International Agency for Research on Cancer (IARC), in 2022, there were approximately 1.4 million new cases of prostate cancer globally, making it the second most frequently diagnosed cancer among men [5]. Notably, prostate cancer is the most commonly diagnosed cancer in males in 118 of 185 countries [6]. These epidemiological trends highlight the urgent need for novel therapeutic strategies to improve treatment outcomes for patients with advanced prostate cancer.

Prostate cancer in its early stages often does not present any noticeable symptoms, which is why it is crucial for men over the age of 50, or those at a higher risk, to undergo regular screenings [7]. However, when the cancer spreads to other parts of the body, such as the bones, it may trigger various signs, including persistent bone pain, fractures, and neurological complications like spinal cord compression [8]. It is essential to recognize that not all individuals with prostate cancer that has spread to the bones will exhibit symptoms, and the severity of these symptoms can differ based on how far the cancer has spread and the specific areas affected. As research indicates, approximately 20% to 30% of patients do not seek medical help until they experience symptoms such as bone pain and general discomfort, which occur after the cancer has already spread to the bones.

Prostate cancer is primarily managed through surgery, radiation therapy, hormone therapy, and chemotherapy. The selection of treatment depends on the tumor stage, patient age, and the risk of recurrence. Since approximately 90% of prostate cancers are androgen-dependent, hormone therapy remains the cornerstone of treatment when the cancer has metastasized to the bone or other organs and is not amenable to surgical interventions. The combination of hormone therapy and chemotherapy aims to suppress androgen levels, thereby reducing tumor stimulation, slowing disease progression, and improving disease control [9,10].

For patients with metastatic castration-resistant prostate cancer (mCRPC) who exhibit resistance to androgen deprivation therapy and present with bone metastases without visceral involvement, the radiopharmaceutical Ra-223 dichloride (Xofigo^®^) has been approved as an effective treatment. Ra-223, an alpha-emitting radionuclide, selectively targets bone metastases, delivering localized radiation to tumor sites while minimizing off-target toxicity. Clinical studies have demonstrated its ability to improve overall survival, delay skeletal-related complications, and enhance the quality of life in mCRPC patients [11,12,13]. However, in most cases, chemotherapy with docetaxel or mitoxantrone remains the only treatment option. Unfortunately, the efficacy of these regimens is suboptimal, often leading to rapid disease progression. Studies have shown that despite the use of chemotherapy, the 5-year survival rate for patients with castration-resistant prostate cancer (CRPC) remains below 30%, with a median survival duration of approximately one year [14,15,16]. This highlights the urgent need for the development of innovative therapeutics with novel mechanisms of action to address the existing treatment gap in CRPC [17,18].

The revolution in personalized medicine has driven the rapid advancement of peptide receptor radionuclide therapy (PRRT), an innovative approach that combines peptide-based drugs with nuclear medicine. Depending on the radiolabeled isotope, peptide-based agents can function as either diagnostic radiopharmaceuticals (e.g., ^68^Ga, ^99^mTc, and ^18^F) or targeted therapeutic agents (e.g., ^177^Lu and ^225^Ac) [19,20,21]. Nuclear medicine molecular imaging is first employed to identify patients expressing the relevant molecular targets. Subsequently, radiolabeled therapeutic agents deliver alpha particles or short-range beta particles directly to tumor cells, enabling precise cancer cell destruction while minimizing damage to surrounding healthy tissues, thereby achieving the goals of personalized medicine [22,23]. Among PRRT agents, the radioisotope Lutetium-177 (^177^Lu) plays a pivotal role in enhancing the efficacy of PRRT treatments. ^177^Lu-based therapies, such as ^177^Lu-DOTATATE for neuroendocrine tumors and ^177^Lu-PSMA for prostate cancer, have demonstrated remarkable therapeutic effects by selectively targeting tumors that overexpress peptide receptors or prostate-specific membrane antigen (PSMA) [24,25,26]. This targeted approach has significantly improved treatment outcomes, reinforcing the importance of PRRT in modern oncology.

Lutetium-177 (^177^Lu) is an emerging radionuclide with significant potential in biomedical applications due to its favorable physical properties. Its emission characteristics align well with lesion size, ensuring precise radiation delivery to cancer cells while minimizing damage to surrounding healthy tissues. With a maximum penetration depth of 2.2 mm and an average depth of 0.67 mm, ^177^Lu is particularly effective for treating small tumors [27,28]. As a medium-energy β-emitter (490 keV, max 0.5 MeV), ^177^Lu enables efficient tumor irradiation. Additionally, it emits low-energy γ-rays (208 keV at 11% and 113 keV at 6%), allowing for ex vivo imaging to facilitate tumor localization and dosimetry assessments [27]. Its relatively long physical half-life (6.73 days) ensures sustained delivery of ^177^Lu-PSMA to prostate cancer cells, enhancing therapeutic efficacy [29]. The γ-ray emissions of ^177^Lu also support single-photon emission computed tomography (SPECT), enabling real-time visualization of radiolabeled peptide distribution and the monitoring of drug clearance post-treatment [21]. ^177^Lu-PSMA-617 therapy specifically targets prostate cancer cells expressing prostate-specific membrane antigen (PSMA), a protein significantly overexpressed in malignant prostate tissue [30,31]. ^177^Lu-PSMA-617 therapy has gained international attention as a promising treatment for Stage 4 prostate cancer. In 2022, the FDA approved ^177^Lu-PSMA-617 (Pluvicto™) for PSMA-positive mCRPC patients who had undergone prior treatments, including androgen receptor pathway inhibitors and taxane-based chemotherapy [32,33].

In this study, we demonstrated the therapeutic efficacy of ^177^Lu-PSMA-617 for bone metastases in a murine model of prostate cancer. We conducted several preclinical evaluations of ^177^Lu-PSMA-617 in tumor-bearing mice, including the development of ^177^Lu radiolabeling and quality control techniques, assessments of radiopharmaceutical stability, cell binding affinity, nuclear imaging, in vivo biodistribution, pharmacokinetics, radiation dosimetry, and antitumor efficacy. To our knowledge, this is the first comprehensive preclinical study to evaluate ^177^Lu-PSMA-617 specifically in a PSMA-positive bone metastasis model that closely mimics the clinical scenario of advanced mCRPC. This integrated approach enables a better understanding of the compound’s tumor-targeting ability, therapeutic impact, and safety in bone-dominant disease. These findings have important implications for the development of treatments for human prostate cancer patients with bone metastases, potentially informing future clinical applications of PSMA-targeted radionuclide therapy.

## 2. Results

### 2.1. Characterization of ^177^Lu-PSMA-617

A precursor amount of 50 nmol was labeled with Lutetium-177, achieving a radiochemical yield of over 99% and a molar activity of 25.5 GBq/μmol within 15 min at 95 °C. The radiochemical purity of ^177^Lu-PSMA-617 (Rt = 7.47 min) exceeded 98%, as determined by radio-HPLC and iTLC analyses (Figure 1). To assess radiolytic stability, ^177^Lu-PSMA-617 was incubated in human serum at 4, 25, and 37 °C for 1, 24, 48, and 72 h. The compound exhibited high stability, with only 1.2% free activity at 4 °C and less than 1.1% at 25 °C after 2 h. Long-term stability assessments showed no detectable free activity after 48 h at all tested temperatures (4, 25, and 37 °C). Additionally, the compound remained stable for up to 72 h at 37 °C without any release of free activity. However, after 72 h, less than 5% and 4% of free ^177^Lu was detected at 4 °C and 25 °C, respectively. The absence of free activity at 37 °C for 72 h indicates that the binding between Lutetium-177 and PSMA-617 remained stable, with no significant dissociation occurring during the stability evaluation. These findings are consistent with other studies [34]. These findings suggest that after an initial minor dissociation, ^177^Lu-PSMA-617 exhibits prolonged stability, retaining its radiolabeling integrity under physiological conditions. Overall, these results underscore the high stability of ^177^Lu-PSMA-617 in human serum, supporting its potential suitability for nuclear medicine applications. The ability to maintain radiolabeling over an extended period reinforces its viability for targeted radiopharmaceutical therapy and molecular imaging in oncology.

### 2.2. Surface Binding and Internalization of ^177^Lu-PSMA-617

In this study, the cell surface binding and internalization of ^177^Lu-PSMA-617 were evaluated in PSMA-positive LNCaP cells and compared to PSMA-negative PC-3 cells (Figure 2). In this study, the uptake in PSMA-positive LNCaP cells demonstrated a clear time-dependent increase in both surface binding and internalization, while no such trend was observed in PSMA-negative PC-3 cells. The surface binding of ^177^Lu-PSMA-617 in PSMA-positive LNCaP cells ranged from 27.7% to 47.9% IA/10^6^ cells following 1 to 4 h of incubation, with an internalized fraction of 2.3% to 13.6% IA/10^6^ cells (Figure 2A). In contrast, PSMA-negative PC-3 cells exhibited minimal surface binding, not exceeding 1.7% IA/10^6^ cells. Furthermore, the internalization of ^177^Lu-PSMA-617 in PC-3 cells was negligible, dropping to <0.1% IA/10^6^ cells (Figure 2B). These results confirm the PSMA-specific surface binding and internalization of ^177^Lu-PSMA-617, highlighting its selective uptake in PSMA-expressing cells. ^177^Lu-PSMA-617 shows specific binding and internalization in PSMA-positive cells, supporting its potential for targeted radionuclide therapy and imaging in prostate cancer [35,36].

### 2.3. Binding Affinity of ^nat^Lu-PSMA-617

As shown in Figure 3, ^177/nat^Lu-PSMA-617 exhibited high affinity for PSMA-positive LNCaP cells. The mean IC_50_ value of ^nat^Lu-labeled PSMA-617 in the competitive inhibition assay was 17.709 nM. Additionally, the *K*_d_ value of ^177^Lu-PSMA-617, determined from saturation binding assays, was 4.358 ± 0.664 nM, consistent with previous reports [37,38]. The binding affinity (*K*_i_) of ^177/nat^Lu-PSMA-617 in LNCaP cells was 7.174 nM, further confirming its high affinity for PSMA-positive cells.

### 2.4. Biodistribution of ^177^Lu-PSMA-617 in Mice with LNCaP Prostate Cancer

An in vivo imaging system was used for bioluminescence imaging (BLI) to visualize ASID mice with bone metastases of LNCaP-luc prostate cancer before the administration of ^177^Lu-PSMA-617 (Figure 4A). Subsequently, ^177^Lu-PSMA-617 (~37 MBq of Lu-177) was administered intravenously to mice with bone metastases, and NanoSPECT/CT imaging was performed at 4, 24, and 48 h post-administration (Figure 4B). Additionally, the mice were sacrificed, and their organs and tissues were extracted for quantitative analysis using a gamma counter at 1, 4, 24, and 48 h after injection (Figure 4C). The results demonstrated that the therapeutic agent specifically bound to bone metastases of prostate cancer, as evidenced by SPECT and bioluminescence imaging. The biodistribution analysis of ^177^Lu-PSMA-617 showed maximum uptake in LNCaP tumors, with 23.31 ± 0.94 %IA/g at 4 h post-injection. The kidneys exhibited a high uptake of 43.83 ± 3.41 %IA/g at 1 h, which gradually decreased over time. Meanwhile, the liver, lungs, and spleen exhibited low uptake, except for the kidneys and urinary bladder.

The tumor-to-normal tissue background (T/N) ratios are depicted in Figure 4D. The tumor-to-blood ratio increased from 69.5 at 1 h post-injection to a maximum of 434.4 at 24 h. The tumor-to-muscle ratio rose from 60.2 at 1 h to a peak of 857.4 at 24 h. The tumor-to-kidney ratio increased from 0.3 at 1 h to a maximum of 26.3 at 48 h. These findings suggest that the therapeutic agent exhibits a high affinity for tumor tissue while demonstrating relatively low uptake in other organs, making it a promising therapeutic option for treating bone metastasis in prostate cancer.

### 2.5. Autoradiography of ^177^Lu-PSMA-617 in Bone Metastasis of Prostate Cancer

To investigate the biodistribution of ^177^Lu-PSMA-617 in bone metastases of prostate cancer following SPECT/CT imaging, we conducted autoradiography studies. Remarkably, autoradiography results clearly demonstrated a substantial accumulation of ^177^Lu-PSMA-617 radioactivity in the tibia containing an LNCaP tumor at 24 h post-injection (Figure 5). Representative autoradiograms further confirmed that ^177^Lu-PSMA-617 directly localizes to PSMA-positive LNCaP prostate cancer cells. This finding definitively establishes the effective targeting of bone metastases by ^177^Lu-PSMA-617, particularly in regions with high PSMA expression, a conclusion further validated by IHC staining. Autoradiographic analysis revealed substantial accumulation of ^177^Lu-PSMA-617-derived radioactivity in the tibia bearing LNCaP tumors at 24 h post-injection, providing strong evidence for the successful targeting of bone metastases. Representative autoradiograms clearly demonstrated that ^177^Lu-PSMA-617 localized specifically to PSMA-positive LNCaP lesions.

### 2.6. Therapeutic Efficacy of ^177^Lu-PSMA-617

Therapeutic response monitoring was performed using bioluminescence imaging on days 3, 7, 15, and 30. Bioluminescence images (Figure 6A) and signal intensity data (Figure 6B) obtained via the IVIS system demonstrated the efficacy of ^177^Lu-PSMA-617 in suppressing the tumor burden in experimental mice compared to the control group. Bioluminescence intensities were plotted on a linear scale against time (days after treatment initiation on day 0), showing significantly lower signal intensities in ^177^Lu-PSMA-617-treated mice compared to controls. The survival curves presented in Figure 6C further highlight the therapeutic effect. The median survival time for the control group (receiving normal saline) was 26 days, whereas mice treated with 1mCi (37 MBq) and 3mCi (111 MBq) of ^177^Lu-PSMA-617 had median survival times of 34 days (*p* = 0.01) and 40 days (*p* = 0.02), respectively. This corresponds to lifespan increases of 30.7% and 53.8%, respectively. These findings indicate that ^177^Lu-PSMA-617 therapy significantly improves tumor suppression and extends survival, demonstrating its potential as an effective radiotherapeutic for prostate cancer bone metastases.

### 2.7. Absorbed Radiation Dose Calculations

Radiation doses (mGy/GBq) for major human organs (male) were calculated using the OLINDA/EXM Version 1.1 software. The organ-specific radiation absorbed doses following the administration of ^177^Lu-PSMA-617 to humans were estimated based on residence times derived from mouse data and are presented in Table 1. The results indicate that certain organs received significant absorbed doses, including the testes (2.16 × 10^−2^ mSv/MBq), lower large intestine (2.01 × 10^−2^ mSv/MBq), stomach wall (1.98 × 10^−2^ mSv/MBq), and red marrow (1.47 × 10^−2^ mSv/MBq). Moderate absorbed doses were observed in organs such as osteogenic cells (4.98 × 10^−3^ mSv/MBq), the skin (1.54 × 10^−3^ mSv/MBq), the urinary bladder wall (8.32 × 10^−3^ mSv/MBq), and the thyroid (8.08 × 10^−3^ mSv/MBq). The overall effective dose was 0.127 mSv/MBq. These findings provide critical insights into the dosimetry profile of ^177^Lu-PSMA-617, aiding in the assessment of its safety and therapeutic potential.

## 3. Discussion

In this study, we successfully synthesized ^177^Lu-PSMA-617 with high radiochemical yield (99.2%), molar activity (25.5 GBq/μmol), and radiochemical purity (>98%). These results are in line with earlier studies [39,40], confirming that established labeling protocols for ^177^Lu-PSMA-617 yield a product of sufficient quality for both preclinical and clinical use. High molar activity is critical for effective receptor saturation with a minimal mass dose, as supported by the work of Eder et al. [41], who demonstrated that increased molar activity correlates with enhanced tumor targeting and reduced off-target effects.

The radiolabeled product exhibited robust in vitro and in vivo stability. Over a 72 h observation period, less than 5% free ^177^Lu was observed even at 37 °C, confirming radiolabeling integrity in physiologic conditions. Comparable findings have been reported by Benesova et al. [42], showing that ^177^Lu-PSMA-617 retains >95% stability in human serum over 96 h. The stability data support its safe pharmacokinetic profile and therapeutic viability, especially for slow-accumulating tumor lesions.

In vitro cell binding studies showed the specific uptake of ^177^Lu-PSMA-617 in PSMA-positive LNCaP cells (up to 47.9% IA/10^6^ cells) and negligible uptake in PSMA-negative PC-3 cells. These observations are consistent with previous reports [40,43] and are further reinforced by internalization rates (up to 13.6% IA/10^6^ cells in LNCaP vs. <0.1% in PC-3), highlighting PSMA-mediated cellular internalization—a prerequisite for effective radioligand therapy [44].

Biodistribution analysis revealed significant tumor accumulation (23.31 ± 0.94 %IA/g at 4 h), which remained relatively high (12.88 ± 0.55 %IA/g) at 24 h. This sustained retention mirrors prior findings by Rahbar et al. [30], who observed that tumor uptake persists at therapeutic levels for over 72 h post-injection. In contrast, renal uptake dropped markedly after the initial high accumulation, indicating efficient clearance and reduced nephrotoxicity risk. Low uptake in the liver, lungs, and spleen is a favorable pharmacokinetic trait and critical for minimizing off-target radiation exposure.

The differential biodistribution and mechanisms of action between ^223^Ra-dichloride (Xofigo™) and ^177^Lu-PSMA-617 underscore fundamental distinctions in their therapeutic profiles. Radium-223 dichloride, an FDA-approved bone-seeking alpha-emitting radiopharmaceutical, accumulates preferentially at bone surfaces adjacent to tumors and in healthy bone rather than within tumor tissue [45]. This localization, driven by bone remodeling, enables it to effectively palliate bone pain and modestly prolong survival in patients with metastatic castration-resistant prostate cancer (mCRPC) [46,47]. However, autoradiographic and histologic evaluations have confirmed that ^223^Ra lacks tumor cell specificity [48], limiting its utility as a monotherapy, particularly in patients with widespread or visceral disease.

In contrast, ^177^Lu-PSMA-617 demonstrates selective and sustained uptake in PSMA-expressing prostate cancer cells, including those within bone lesions, as confirmed by autoradiographic and immunohistochemical analyses. Unlike ^223^Ra, which targets the bone microenvironment, ^177^Lu-PSMA-617 delivers therapeutic radiation directly to tumor cells, providing a more precise and mechanistically favorable approach [48]. This tumor specificity reduces off-target radiation exposure to healthy bone marrow, potentially lowering hematologic toxicity, and enables integration with PSMA-targeted imaging agents (e.g., ^68^Ga-PSMA or ^89^Zr-PSMA) in a theranostic framework. Despite their differences, the complementary mechanisms of ^223^Ra and ^177^Lu-PSMA-617 suggest potential value in combination or sequential treatment strategies. For example, ^223^Ra’s bone remodeling-based targeting could be used to modulate the tumor microenvironment, while ^177^Lu-PSMA-617 directly eradicates tumor cells. However, such combination regimens require careful evaluations to avoid overlapping toxicities, particularly to the bone marrow.

Additionally, the broader biodistribution of ^177^Lu-PSMA-617 allows it to address both skeletal and soft tissue (e.g., nodal and visceral) metastases, thereby offering a more comprehensive therapeutic reach that aligns with the heterogeneous metastatic patterns of advanced mCRPC. The beta-emitting nature of ^177^Lu also confers a crossfire effect, which may help overcome intratumoral heterogeneity and target adjacent PSMA-low tumor cells. These characteristics, together with the agent’s tumor specificity and favorable pharmacokinetic profile, underscore its potential clinical utility. In summary, the accumulated evidence underscores the clinical promise of ^177^Lu-PSMA-617 as a highly versatile and tumor-specific radioligand capable of effectively targeting both skeletal and visceral metastases in advanced prostate cancer [45,46,47,48]. Its combined diagnostic and therapeutic functionality, coupled with its broader disease coverage, represents a substantial advancement over conventional bone-targeting agents. These attributes highlight the need for further clinical investigation to refine its optimal dosing strategies, treatment sequencing, and long-term safety across diverse patient populations.

Therapeutic efficacy studies have demonstrated that ^177^Lu-PSMA-617 produces dose-dependent improvements in survival, with extensions of 30.7% and 53.8% observed at 37 and 111 MBq, respectively. Bioluminescence imaging confirmed reduced tumor burden in treated groups. These results are consistent with previous preclinical studies. It has been reported that significant tumor regression and prolonged survival using high-specific-activity ^177^Lu-PSMA-617 are seen in a syngeneic murine prostate cancer model, with superior efficacy observed at higher specific activities and no subacute hematologic toxicity [49]. Similarly, a single 111 MBq dose of ^177^Lu-PSMA-617 has been shown to significantly inhibit tumor growth and prolong median survival to 130 days in mice bearing PSMA-positive tumors [50]. Collectively, these findings reinforce the robust antitumor activity and therapeutic potential of ^177^Lu-PSMA-617 in targeting PSMA-expressing prostate cancer, supporting its continued development for clinical application.

From a dosimetric perspective, estimated radiation doses in major organs using the OLINDA/EXM Version 1.1 software demonstrated a favorable distribution, with rapid urinary clearance (>90% within 24 h). These estimates are consistent with previously reported clinical data [51], showing that the kidneys and salivary glands receive the highest absorbed doses, while marrow and liver exposure remain low—an important safety feature for repeated dosing. Another study investigated the in vivo therapeutic efficacy of ^177^Lu-PSMA-617 in mice bearing PSMA-positive tumors. The study found that a single intravenous dose of 111 MBq of ^177^Lu-PSMA-617 significantly suppressed tumor growth and improved survival compared to untreated controls. Kaplan–Meier survival analysis showed a median survival duration of 130 days for the ^177^Lu-PSMA-617-treated group, highlighting its potent antitumor activity [50]. These consistent results across studies reinforce the therapeutic potential of ^177^Lu-PSMA-617 in targeting PSMA-expressing prostate cancer, providing a strong rationale for its continued development and clinical application.

In addition to its demonstrated efficacy, ^177^Lu-PSMA-617 holds considerable potential for broader clinical application. As a theranostic agent, it may be integrated with diagnostic imaging modalities such as ^68^Ga-PSMA PET to enable patient selection, individualized treatment planning, and real-time monitoring of the therapeutic response. Its ability to target both skeletal and visceral lesions also suggests its value in systemic therapy for advanced mCRPC, particularly in cases refractory to hormone or chemotherapy. Furthermore, its favorable pharmacokinetic and safety profile supports the potential for repeated dosing and inclusion in combination regimens with other systemic or immune-based therapies. These applications underscore the relevance of our findings and support future clinical translation of ^177^Lu-PSMA-617 as a targeted radiopharmaceutical for treating skeletal or visceral metastases in advanced prostate cancer.

In summary, our findings suggest that ^177^Lu-PSMA-617 holds considerable promise as a targeted radiopharmaceutical, with properties that support both therapeutic efficacy and safety. Its favorable pharmacological profile and selective uptake in PSMA-positive lesions distinguish it from other agents such as ^223^Ra [11]. Importantly, these preclinical outcomes provide a strong rationale for advancing this compound toward clinical trials, particularly for patients with advanced-stage prostate cancer exhibiting skeletal and visceral metastases.

## 4. Materials and Methods

### 4.1. Preparation of ^177^Lu-PSMA-617

Radiolabeling was conducted as previously described [30]. A total of 1 GBq of non-carrier added lutetium-177 (Isotopia Molecular Imaging, Petach Tikva, Israel) was mixed with 100 μL of a sodium acetate solution (0.4 M, pH 5.5), 1.25 μL of an ascorbic acid solution (20% *w*/*w*), and 5 μL of a PSMA-617 solution (10 mM, Ontores Biotechnology Co., Ltd., Hangzhou, China). The mixture solution was subjected to heating at a temperature of 95 °C while being placed on a shaker for 15 min. The labeling efficiency was measured using instant thin-layer chromatography (iTLC) on the glass microfiber chromatography paper impregnated with a silica gel (Agilent Technologies, Santa Clara, CA, USA), where 100 mM EDTA in 100 mM ammonium acetate acts as a mobile phase as described previously [27]. The iTLC sheets were measured using a radioactive scanner (AR-2000 radio-TLC Imaging Scanner, Bioscan, Evreux, France). The labeling rate of >90% was acceptable as performed in the in vitro and in vivo studies. Radiochemical purity was also analyzed by high-performance liquid chromatography (HPLC) with a UV detector (220 nm) and a radio detector using a Zorbax SB-C18 column (Agilent Technologies, Santa Clara, CA, USA). The flow rate was 0.8 mL/min with the gradient mobile phase going from the 80% A buffer (0.1% TFA in water) and 20% B buffer (0.1% TFA in acetonitrile) to the 40% B buffer within 20 min. A sterile 0.9% sodium chloride solution was added before injections in mice. The radiochemical stability of ^177^Lu-PSMA-617 in human serum up to 96 h at 4, 25, and 37 °C was tested by iTLC.

### 4.2. Surface Binding and Internalization of ^177^Lu-PSMA-617

The LNCaP (expressed a higher level of PSMA) human prostate cancer cell line was purchased from the American Type Culture Collection (ATCC) (Manassas, VA, USA) (ATCC # CRL-1740) for cell viability and cellular uptake studies. LNCaP cells were maintained in RPMI 1640 (SIGMA-Aldrich, St. Louis, MO, USA) with 10% fetal bovine serum (FBS; GIBCO, Grand Island, NY, USA) and 1% P/S (Pen Strep; GIBCO, Grand Island, NY, USA) and were incubated at 37 °C with 5% CO_2_. Briefly, 10^5^ LNCaP cells were incubated for 1, 2, and 4 h with ^177^Lu-PSMA-617 (1.5 nM) at 37 °C. The surface-bound radioactivity was removed with glycine HCl (50 mM; pH 2.8) and the internalized fraction using 1 N NaOH. The collected fractions were measured in a gamma counter and calculated as percentage initial activity per 10^6^ cells (%IA/10^6^ cells).

Cell surface binding and internalization experiments were performed with ^177^Lu-PSMA-617 using PSMA-positive LNCaP cells and PSMA-negative PC-3 cells in order to determine if the uptake of ^177^Lu-PSMA-617 was PSMA-specific. For this purpose, cells were seeded in 24-well plates (10^5^ cells in 1 mL of the RPMI medium/well), allowing adhesion and growth overnight at 37 °C. After the removal of the supernatant, cells were washed once with PBS prior to the addition of the RPMI medium without supplements, followed by the addition of the radiolabeled ^177^Lu-PSMA-617 (in MBq/nmol) to each well. The well plates were incubated at 37 °C for 1, 2, and 4 h, respectively. The cells were washed three times with ice-cold PBS to determine the total uptake of ^177^Lu-PSMA-617. The internalized fraction of ^177^Lu-PSMA-617 was determined in cells which were washed with ice-cold PBS, then incubated for 10 min with acidic stripping buffer (50 mM glycine, pH 2.8) followed by an additional washing step with ice-cold PBS. Cell samples were lysed by the addition of NaOH (1 N, 1 mL) to each well. The samples of the cell suspensions were measured in a γ-counter (PerkinElmer, Waltham, MA, USA).

### 4.3. Binding Affinity of ^nat^Lu-PSMA-617

The competitive inhibition assay was performed in 10^5^ LNCaP cells/well as previously described [41]. The radioligand (^177^Lu-PSMA-617) was added as a 0.75 nM solution to 10 different concentrations of non-labeled, ^nat^Lu-labeled PSMA-617 (from 0.01 to 10,000 nM). After 1 h of incubation at 4 °C, the cells were washed with ice-cold PBS, and the cell-bound radioactivity was measured in a gamma counter (PerkinElmer, Waltham, MA, USA). The IC_50_ (half-maximal inhibitory concentration) value was determined using GraphPad Prism version 5.0 (Graph Pad Software, San Diego, CA, USA). Determination of the dissociation constants (*K*_d_ values) was performed by saturation binding assays using LNCaP cells [37,38]. A total of 10^5^ cells were seeded in 96-well plates (cells in 100 µL of the RPMI medium/well), allowing adhesion and growth overnight at 37 °C and 5% CO_2_. After the removal of the supernatant, the cells were washed once with ice-cold PBS at pH of 7.4 prior to the addition of different concentrations of ^177^Lu-PSMA-617 (2.5–100 nM) in the ice-cold RPMI medium without supplements. The cells were incubated for 1 h at 4 °C. Then, the supernatants were removed, and the cells were washed twice with ice-cold PBS followed by the addition of NaOH (1 M, 600 µL) to each well. The cell suspensions were transferred to tubes for measurement in a γ-counter. The *K*_d_ values were determined by plotting specific binding (total binding minus unspecific binding) against the molar concentration of the added radioligands followed by nonlinear regression analysis using the GraphPad Prism 5 software. The binding affinity (*K*_i_) of ^177/nat^Lu-PSMA-617 was calculated using the following formula as described previously [52]: *K*_i_ = IC_50_/[1 + ([radioligand]/*K*_d_)].

### 4.4. Bone Metastasis of Prostate Cancer

Advanced severe immunodeficiency (ASID, NOD.Cg-*Prkdc^scid^Il2rg^tm1Wjl/^*YckNarl) mice were purchased from the National Laboratory Animal Center (Taipei, Taiwan), maintained in a barrier facility, and allowed free access to food and water. All animal procedures were performed following approved protocols that were developed in accordance with recommendations for the proper use and care of laboratory animals at the Institute of Nuclear Energy Research (approved number: 110009). The mice were kept at 21–23 °C at a light–dark cycle of 12 h.

LNCaP-luc cells were harvested from culture and resuspended to 2 × 10^7^ cells/mL in PBS. The cells were combined with an equal volume of Matrigel (BD Biosciences, Bedford, MA, USA) such that 5 × 10^5^ cells were prepared for injection into the right tibia of anesthetized (inhaled anesthetic of 1–2% isoflurane) ASID mice. Intratibial injections were performed using a 29-gauge, 0.5-inch needle inserted through the tibial plateau of the flexed knee as described previously [53,54]. The tumor was imaged, detected, and measured using a Bruker Small Animal Optical Imaging System (In Vivo Xtreme; Billerica, MA, USA). Mice were anesthetized with a mixture of oxygen and isoflurane, then intraperitoneally injected with 100 μL of D-luciferin (Xenogen; 30 mg/mL in PBS).

### 4.5. Biodistribution of ^177^Lu-PSMA-617

Mice received an intravenous injection of ^177^Lu-PSMA-617 (equivalent to 37 MBq of ^177^Lu) at 45 days after tumor implantation. The NanoSPECT/CT (Mediso Medical Imaging Systems, Arlington, VA, USA) was used to image in vivo. After the administration of ^177^Lu-PSMA-617, mice inhaled an anesthetic containing 1–2% isoflurane during imaging acquisition. SPECT and X-ray CT images were obtained at 1, 4, 24, and 48 h after injection. NanoSPECT imaging was acquired using nine multipinhole gamma detectors and high-resolution collimators. The energy window was set to 176, 384 and 497 keV ± 10%; the image size was set to 256 × 256, and the field of view of 60 mm × 100 mm was used. Mice received an intravenous injection of ^177^Lu-PSMA-617 (equivalent to 37 MBq of ^177^Lu) at 45 days after tumor implantation. The distribution of ^177^Lu-PSMA-617 in the mice bearing tumors was evaluated by a gamma counter (PerkinElmer, Waltham, MA, USA). The mice were sacrificed by carbon dioxide euthanasia at 1, 4, 24, and 48 h after administration. The blood, tumor, and normal organ/tissue were collected, and the uptake of radioactivity was measured by a gamma counter. The distribution data obtained using radioactivity count methods are plotted as %ID/g.

### 4.6. Autoradiography of ^177^Lu-PSMA-617 in Bone Metastasis of Prostate Cancer

After the biodistribution studies, the legs and tumors of the mice were frozen and embedded in the optimal cutting temperature compound (Tissue Tek, Sakura, Torrance, CA, USA) for the frozen sections. The frozen sections were placed in a BASMS 2040 imaging plate (Fujifilm, Tokyo, Japan) for autoradiography, and the images were analyzed by an FLA-5100 reader (Fujifilm, Tokyo, Japan) and Multi Gauge V3.0 software (Fujifilm, Tokyo, Japan) [55].

### 4.7. Immunohistochemical Analysis

In order to confirm ^177^Lu-PSMA-617 accumulated in the bone metastasis of the LNCaP tumor, immunohistochemical staining of PSMA and H&E staining were performed. After the mice were sacrificed, the legs and tumors were excised and weighed. For immunohistochemical and H&E staining procedures, the tissues were fixed in formalin and stained with hematoxylin and eosin (H&E). The sections were stained immunohistochemically for PSMA (prostate-specific membrane antigen) expression, treated with 3% hydrogen peroxide for 15 min to quench endogenous peroxidase activity, blocked with 10% normal goat serum for 15 min, rinsed 3 times with PBS for 2 min, incubated overnight at 4 °C with antibodies specific for PSMA (ab19071; Abcam, Cambridge, UK; 1:1000 dilution), rinsed again with PBS, incubated with biotinylated secondary antibodies for 30 min at room temperature, incubated with an avidin–biotin complex, and visualized by the addition of the 3,3′-diaminobenzidine tetrahydrochloride (DAB) chromogen. Immunostaining was carried out using the Histostain-Plus Kit (Zymed Laboratories, Inc., San Francisco, CA, USA).

### 4.8. Therapeutic Efficacy of ^177^Lu-PSMA-617

The mice were intratibially injected with LNCaP/Luc cells. After 45 days of tumor-implantation, the mice were assigned to three groups, including the saline control (*n* = 6), a low dose of ^177^Lu-PSMA-617 (*n* = 8), and a high dose of ^177^Lu-PSMA-617 (*n* = 9). The mice were administered via the tail vein with equivalent to 1mCi (37 MBq) and 3mCi (111 MBq) of ^177^Lu-PSMA-617 on day 0. Body weight changes were measured during observation days. The death of mice was recorded for the survival curve by the Kaplan–Meier method. A humane endpoint was defined as a decrease in body weight of 20% or more compared with the body weight measured on the day of tumor inoculation. The survival rate of all groups was analyzed by the log-rank test, and *p* < 0.05 was considered significant difference. The median survival time was presented in efficacy studies. The median survival time was calculated as the smallest survival time for which the survivor function is less than or equal to 0.5. The therapeutic responses were monitored by bioluminescence imaging during observation days. The BNL/Luc tumor of mice was imaged and measured using the IVIS Imaging system and Living Imaging software (Xenogen, Alameda, CA, USA; available at https://www.perkinelmer.com) on days 3, 7, 15, and 30.

### 4.9. Absorbed Radiation Dose Calculations

The absorbed radiation dose calculations were performed as previously reported [56]. For the estimation of mean absorbed doses in humans, the relative organ mass scaling method was used. The calculated mean value of the percentage injected activity per gram of tissue (%IA/g) for the organs in mice was extrapolated to the uptake amount in the organs of a 70 kg adult using the following formula:[(%IA/g_organ_)_animal_ × (kg_TB weight_)_animal_] × (g_organ_/kg_TB weight_)_human_ = (%IA/organ)_human_

The extrapolated values (%IA) in human organs at 1, 4, 24, and 48 h were fitted with exponential biokinetic models and integrated to obtain the number of disintegrations in the source organs; this information was entered into the OLINDA/EXM computer program. The integrals (MBq-s) for 11 organs, including heart contents (blood), the heart, the liver, the spleen, the lung, the kidney, muscle, the brain, bone, the small intestine, and the remainder of the body, were evaluated and used for the dosimetry evaluation.

### 4.10. Statistical Analysis

All quantitative data are presented as the mean ± standard deviation (SD) unless otherwise specified. Statistical analysis of survival data was performed using the log-rank (Mantel–Cox) test to compare Kaplan–Meier survival curves between the treatment and control groups. A *p*-value of <0.05 was considered statistically significant. GraphPad Prism version 5.0 (GraphPad Software, San Diego, CA, USA) was used for all statistical analyses.

## 5. Conclusions

This study provides a comprehensive preclinical evaluation of ^177^Lu-PSMA-617 in a clinically relevant mouse model of prostate cancer with bone metastases. The radiopharmaceutical demonstrated high radiochemical purity, strong PSMA-specific binding, favorable pharmacokinetics, and selective accumulation in PSMA-positive bone lesions. Therapeutic studies confirmed dose-dependent tumor suppression and survival extension, while radiation dosimetry analyses indicated a favorable safety profile. These findings highlight the dual imaging and therapeutic potential of ^177^Lu-PSMA-617 and support its continued development for the treatment of metastatic castration-resistant prostate cancer, particularly in cases involving bone and visceral metastases.

## Figures and Tables

**Figure 1 ijms-26-05970-f001:**
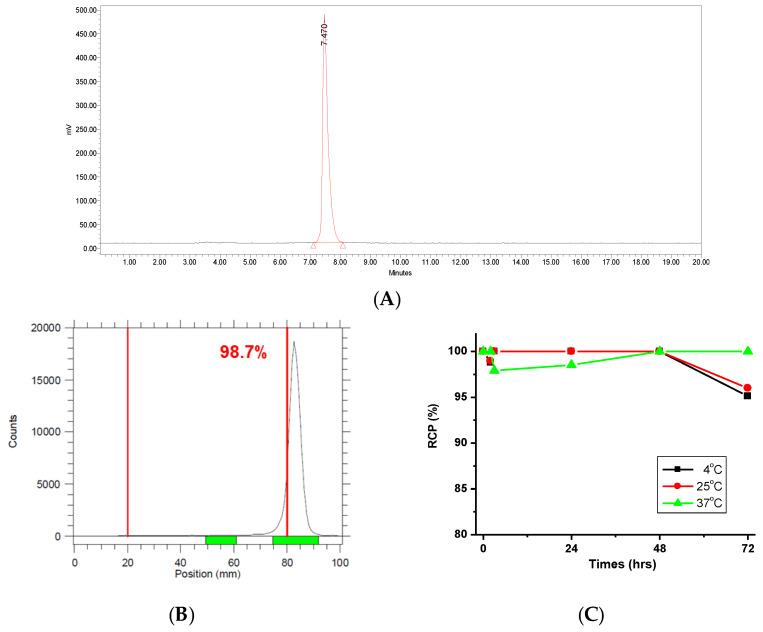
Radiochemical purity (RCP, %) of ^177^Lu-PSMA-617 was measured by (**A**) radio-HPLC and (**B**) iTLC. (**C**) The radiochemical stability of ^177^Lu-PSMA-617 in human serum.

**Figure 2 ijms-26-05970-f002:**
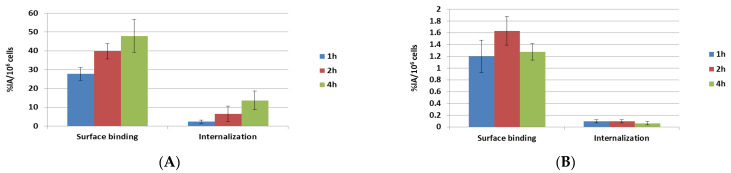
The cell surface binding and internalization of ^177^Lu-PSMA-617 in (**A**) PSMA-positive LNCaP cells were compared to those in (**B**) PSMA-negative PC-3 cells. Results are presented as %IA per 10^6^ cells (mean ± SD, n = 3).

**Figure 3 ijms-26-05970-f003:**
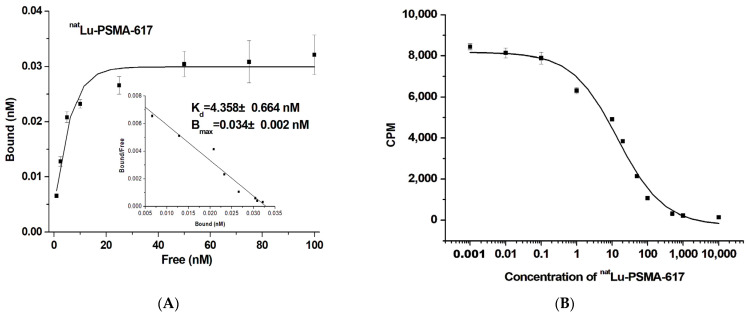
(**A**) Saturation binding study and (**B**) competitive inhibition assay of ^177/nat^Lu-PSMA-617 on intact LNCaP cells. Increasing concentrations of ^nat^Lu-PSMA-617 were used, ranging from 0.01 to 10,000 nM. All radiotracers exhibited high affinity for the PSMA-positive LNCaP cells. The dissociation constant (*K*_d_) and the maximum number of binding sites (*B*_max_) were calculated from nonlinear regression analysis using GraphPad Prism. Data are presented as counts per minute (CPM).

**Figure 4 ijms-26-05970-f004:**
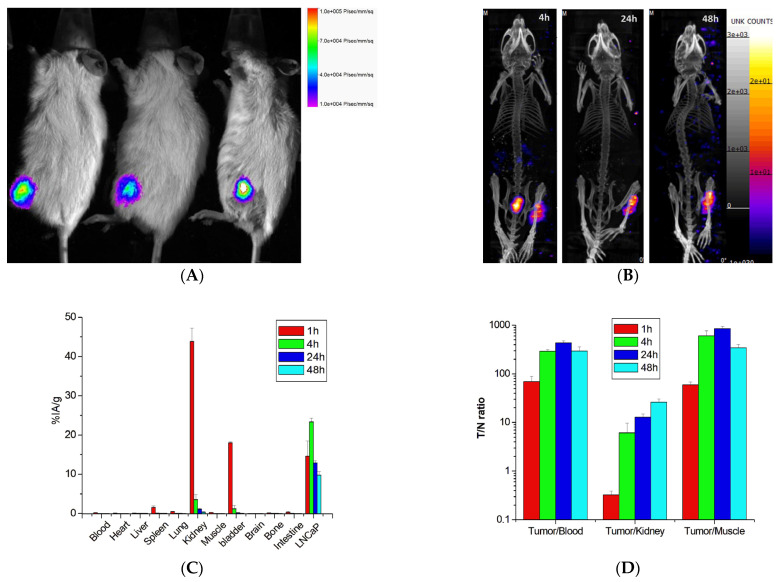
(**A**) Bioluminescence imaging of ASID mice with bone metastasis of LNCaP-luc prostate cancer, (**B**) SPECT/CT images, and (**C**) biodistribution of ^177^Lu-PSMA-617 (%IA/g: percentage of injected activity per gram of tissue) in mice after intravenous injection (n = 3 at each time point). (**D**) Tumor-to-normal tissue (T/N) ratios of ^177^Lu-PSMA-617.

**Figure 5 ijms-26-05970-f005:**
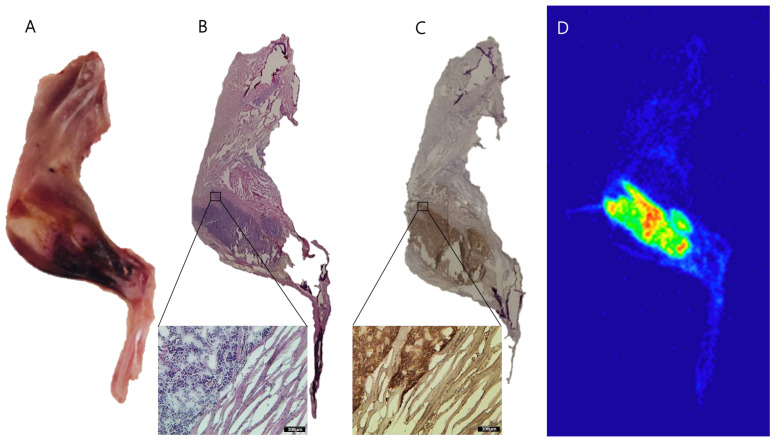
Undecalcified whole-mount cryosection and autoradiography of intratibial LNCaP prostate cancer xenograft models. (**A**) En bloc color macrophotograph of an embedded sample at the cutting surface. (**B**) Hematoxylin and eosin (H&E) stain of the whole-mount adhesive bound section. (**C**) Immunohistochemistry with the anti-PSMA antibody. Insets are images of the respective stained section of the tumor. Scale bars, 100 μm. (**D**) Autoradiography of ^177^Lu-PSMA-617 and daughter emission. Areas of intense uptake colocalize with active bone metastasis sites in the distal femur and proximal tibia.

**Figure 6 ijms-26-05970-f006:**
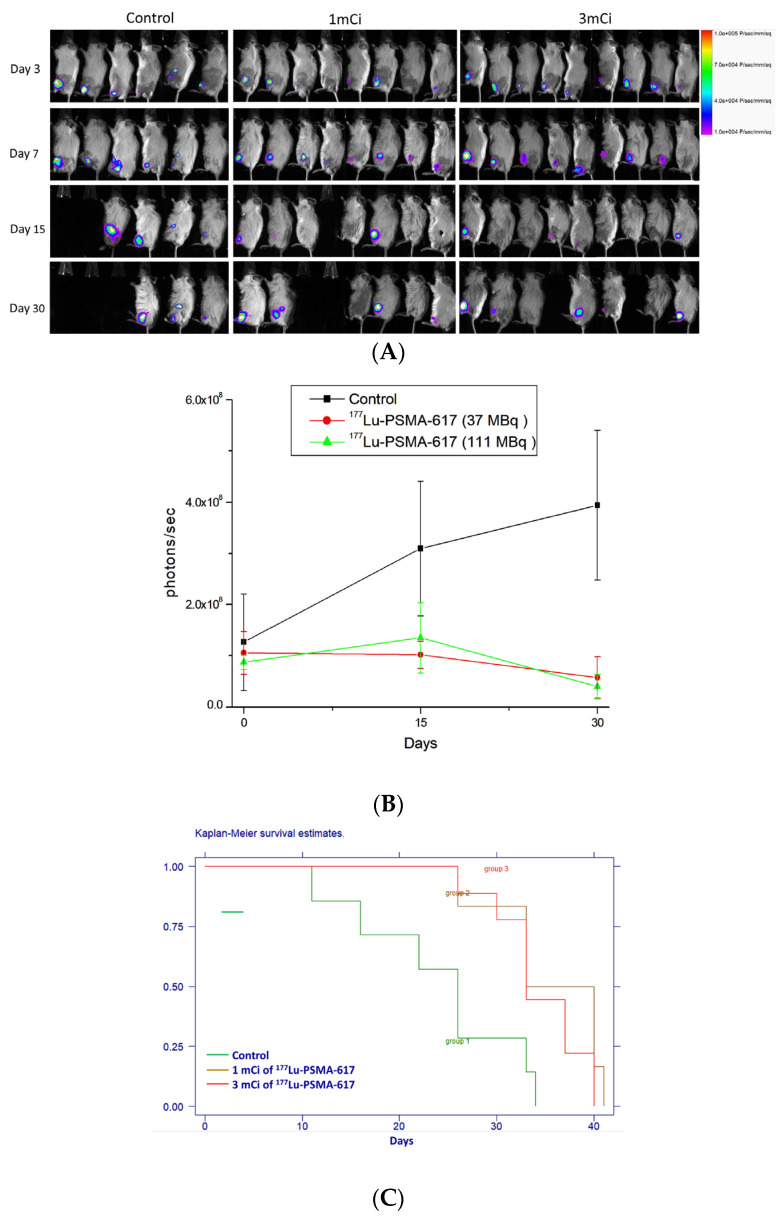
The antitumor efficacy of ^177^Lu-PSMA-617 in mice with bone metastasis of prostate cancer. (**A**) Time-lapse bioluminescence imaging of mice bearing LNCaP/Luc prostate cancer after the treatment with saline (control), 1mCi of ^177^Lu-PSMA-617 (*n* = 8), and 3mCi of ^177^Lu-PSMA-617 (*n* = 9). (**B**) Quantification of bioluminescence in (**A**). (**C**) Kaplan–Meier survival curves of mice with bone metastasis of prostate cancer after treatment. The survival rate data comparisons were performed using the log-rank (Mantel–Cox) test; *p* < 0.05 was compared with the control.

**Table 1 ijms-26-05970-t001:** Radiation dose estimates for ^177^Lu-PSMA-617 in humans.

Organ	Estimated Dose (mSv/MBq)
Adrenals	8.15E−04
Brain	3.54E−05
Breasts	7.86E−03
LLI Wall	2.01E−02
Small Intestine	8.52E−04
Stomach Wall	1.98E−02
ULI Wall	8.45E−04
Kidneys	2.19E−04
Liver	5.81E−04
Lungs	1.47E−03
Muscle	6.33E−05
Pancreas	8.27E−04
Red Marrow	1.47E−02
Osteogenic Cells	4.98E−03
Skin	1.54E−03
Spleen	6.92E−05
Testes	2.16E−02
Thymus	8.01E−04
Thyroid	8.08E−03
Urinary Bladder Wall	8.32E−03
Uterus	8.48E−04
Effective Dose	1.27E−01

Dosimetry was analyzed by the OLINDA/EXM Version 1.1 software. Extrapolated radiation dose for a 70 kg male adult. LLI, lower large intestine; ULI, upper large intestine.

## Data Availability

The original contributions presented in this study are included in the article. Further inquiries can be directed to the corresponding author.

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
