# Peer review of "Exploring the Therapeutic Potential of ^177^Lu-PSMA-617 in a Mouse Model of Prostate Cancer Bone Metastases"

_ijms, 2025, doi:10.3390/ijms26135970_

Round 1

Reviewer 1 Report

Comments and Suggestions for Authors

The authors evaluated the therapeutic potential of 177Lu-PSMA-617 in prostate bone metastasis by the intratibial xenograft mouse model.

Some comments are listed below.

  1. Line 108: Texts need to be rechecked and revised.
  2. Lines 215-216: Make sure the procedure follows the IACUC (Institutional Animal Care and Use Committee) regulation and approved animal protocols.
  3. Figure 1 (C): The authors should provide the full name of RCP (%) in the legend. The Y axis between 90-100% could enlarge to demonstrate the difference among the 4, 25, and 37° C groups.
  4. Figures 2 (a) and (b): The X-axis data is not complete. What are the meanings of different colors? Are there any significant differences between groups?
  5. Line 304: ¹⁷⁷/natLu-PSMA-617 or ¹⁷⁷/natLu-PSMA-617 should be consistent.
  6. Figure 3: (a) missing the 100nM at the X axis. (b) Provide the full name of CPM in the figure legend.
  7. Lines 314-316 and 325: Figure 1A? Figure 1B? Figure 1D?
  8. Figures 4 (c) and (d): Any significant difference among the groups or between groups? %IA/g and T/N ratio: list full name in the figure legend.
  9. Figure 5: (B) and (C) provide high-fold images of HE and IHC staining to visualize the cell morphology and positive staining cells.
  10. Figure 6: (a) left side labels and images need to be corrected following the order of days 3, 7, 15, and 30. The cancer cells of the control and 3mCi groups of the first mouse from the right side seem not to have been detected by IVIS through days 3, 7, 15, and 30, which may indicate that the intratibial injection or cancer cell incubation failed. Please recheck the Day 0 images to ensure the tumor cells have been successfully incubated in the mouse. If not, these mice may need to be excluded from the study. (b) and (c): Are there any significant differences between groups? N number of controls should be provided in the figure legend.
  11. The discussion section suggests including a comparison of ¹⁷⁷Lu-PSMA-617 with recently published targeted therapies for prostate cancer bone metastasis.
  12. Include statistic methods in the method section.

Reviewer 2 Report

Comments and Suggestions for Authors

The work entitled “Exploring the Therapeutic Potential of 177Lu-PSMA-617 in a Mouse Model of Prostate Cancer Bone Metastases” investigated the therapeutic potential of ¹⁷⁷Lu-PSMA-617, a PSMA-targeted radiopharmaceutical, in a murine model of prostate cancer bone metastases. The results highlight ¹⁷⁷Lu-PSMA-617 as a promising targeted radiotherapeutic agent for both bone and visceral metastases in prostate cancer, with substantial potential for clinical application in molecular imaging and targeted therapy. Besides the encouraging results, the data is sufficient and solid throughout the manuscript. It is recommended to accept the manuscript by International Journal of Molecular Science after addressing the concerns below.

#1 Please clarify the work novelty. Please clearly articulate the novelty of the study in both the Abstract and the last paragraph of the Introduction section.

#2 Please polish the figures used in this manuscript. For instance, the text size within Figure 1 needs to be unified. Figure 2 needs colour describing information. Also, Figure 2 (a) needs to be modified. Similarly, please optimize the figures 3 to 6, accordingly. To promote the figure quality, it is recommended to find a work with a high impacted factor for reference.

#3 Please add a conclusion section for this work. There is no conclusion section for this work. Please add a conclusion section for this manuscript.

#4 Please find 10 to 15 relevant studies for discussion. The current Discussion section is structured according to the sequence of figures, and much of the content largely reiterates the results without sufficient analytical depth. It is recommended that the authors incorporate 10 to 15 relevant studies to support a more comprehensive discussion. A lateral comparison between the findings of previous research and those presented in this study is essential. The authors are encouraged to focus the discussion on such comparative analysis to better contextualize the novelty and significance of their results.

#5 Please talk about the potential applications of the 177Lu-PSMA-617. Please provide several sentences in the discussion section to illustrate the potentials applications of the drug in this research.  

Comments on the Quality of English Language

The work is clearly presented regarding English writing.

Round 2

Reviewer 1 Report

Comments and Suggestions for Authors

In Figure 6 (a), the image panel has not been corrected to follow the chronological order: Day 3, 7, 15, and 30.

Author Response

Comments1: In Figure 6 (a), the image panel has not been corrected to follow the chronological order: Day 3, 7, 15, and 30.

Response 1: Thank you for your valuable comment. We have corrected the image panel in Figure 6(a) to follow the proper chronological order: Day 3, 7, 15, and 30. The revised figure has been updated in the manuscript accordingly.

Reviewer 2 Report

Comments and Suggestions for Authors

Please check the whole manuscript before publication.

Author Response

Comments 1: Please check the whole manuscript before publication.

Response1: Thank you for your important suggestion. In response, we have carefully re-examined the entire manuscript and identified and corrected multiple typographical, grammatical, and formatting inconsistencies. These include subject-verb agreement, standardized spacing between numerals and units, consistent use of SI units (e.g., μL instead of uL), and harmonization of terminology (e.g., PSMA-positive vs. PSMA+). We appreciate your feedback, which has helped us enhance the overall clarity and accuracy of the manuscript.